# The Role of Epicardial Adipose Tissue in Acute Coronary Syndromes, Post-Infarct Remodeling and Cardiac Regeneration

**DOI:** 10.3390/ijms25073583

**Published:** 2024-03-22

**Authors:** Kamil Krauz, Marcel Kempiński, Paweł Jańczak, Karol Momot, Maciej Zarębiński, Izabela Poprawa, Małgorzata Wojciechowska

**Affiliations:** 1Chair and Department of Experimental and Clinical Physiology, Laboratory of Centre for Preclinical Research, Medical University of Warsaw, Banacha 1b, 02-097 Warsaw, Poland; kamilkrauz01@gmail.com (K.K.); marckemp2002@gmail.com (M.K.); pawel49j@gmail.com (P.J.); karol.momot@wum.edu.pl (K.M.); 2Department of Invasive Cardiology, Independent Public Specialist Western Hospital John Paul II, Lazarski University, Daleka 11, 05-825 Grodzisk Mazowiecki, Poland; maciej@zarebinski.pl (M.Z.); izapoprawa@gmail.com (I.P.)

**Keywords:** epicardial adipose tissue, acute coronary syndromes, coronary artery disease, myocardial infarction, left ventricular remodeling, post-infarction heart failure, stem cells

## Abstract

Epicardial adipose tissue (EAT) is a fat deposit surrounding the heart and located under the visceral layer of the pericardium. Due to its unique features, the contribution of EAT to the pathogenesis of cardiovascular and metabolic disorders is extensively studied. Especially, EAT can be associated with the onset and development of coronary artery disease, myocardial infarction and post-infarct heart failure which all are significant problems for public health. In this article, we focus on the mechanisms of how EAT impacts acute coronary syndromes. Particular emphasis was placed on the role of inflammation and adipokines secreted by EAT. Moreover, we present how EAT affects the remodeling of the heart following myocardial infarction. We further review the role of EAT as a source of stem cells for cardiac regeneration. In addition, we describe the imaging assessment of EAT, its prognostic value, and its correlation with the clinical characteristics of patients.

## 1. Introduction

Epicardial adipose tissue (EAT) is the fat deposit situated between the myocardium and the visceral layer of the pericardium. Histologically, it is considered white but may exhibit beige or brown features. EAT is composed of adipocytes, ganglia, interconnecting neurons, stromovascular, and immune cells [1,2,3,4]. It also contains progenitor cells that could be transformed into myofibroblasts [5]. EAT is anatomically and functionally contiguous with the myocardium and coronary vessels [4,6], covers 80% of the heart’s surface, and contributes to 20% of the heart’s weight [7].

As a part of EAT, pericoronary adipose tissue (PCAT) is a fat deposit that surrounds coronary vessels and is adjacent to the adventitia of coronary arteries (Figure 1) [8,9]. That differentiation is important because EAT and PCAT could contribute to cardiovascular risk in distinct ways, giving different pathogenic effects [8,10]. It is also vital to distinguish EAT from pericardial adipose tissue (PAT) as they have different anatomical locations and functional properties [11].

Physiologically EAT displays metabolic and thermogenic properties. It also provides mechanical protection of the heart and coronary arteries [4]. Additionally, EAT appeared to be an endocrine organ, and as a rich source of adiponectin, adrenomedullin, resistin, and many other cytokines [6,12] may have pro- or anti-inflammatory effects on the heart [12,13]. Communication between EAT and myocardium can be achieved via paracrine and vasocrine signaling [4,14].

Due to its unique structure and features, the role of EAT has drawn researchers’ attention in terms of metabolic and cardiovascular (CV) disorders [15], including coronary artery disease (CAD) [16,17], heart failure (HF) [18], or atrial fibrillation (AF) [19]. Most research concerns the relationship between adipose tissue and atherosclerosis. Studies show that the paracrine activity of EAT may vary and, in certain circumstances, may contribute to the development of atherosclerotic plaque formation and its rupture [4,14,20]. Since it would be difficult to assess the metabolic activity of EAT, a good solution seems to be the assessment of its amount. It is not fully explored, however, whether the volume of EAT assessed by imaging studies may be positively associated with CV disorders [4,21,22,23].

As most studies focus on chronic coronary syndromes, this article aims to review associations between EAT and acute coronary syndromes (ACS) and to discuss possible mechanisms through which EAT contributes to cardiac remodeling following myocardial infarction (MI). The potential role of EAT as a source of stem cells, as well as imaging assessment, was also described.

## 2. The Contribution of Epicardial Adipose Tissue to Pathogenesis of Acute Coronary Syndromes

Various biological processes occurring in EAT may impact the myocardium, as they share a common microvasculature and are located in close proximity. EAT plays a role in the development of CAD which is the main underlying cause of ACS [14]. Several studies analyzed whether and how EAT contributes to the pathogenesis of ACS, therefore this topic requires to be summarized (Figure 2) (Table 1). Special attention was given to the role of inflammation and adipokines secreted by EAT.

The most common mechanism leading to MI is the rupture of a vulnerable plaque and thrombotic occlusion of the coronary artery [24]. Cellular crosstalk between EAT and fibroatheromas may contribute to the development of their high-risk characteristics [25]. Thinning of the fibrous cap and necrotic core development may result from the apoptosis of smooth muscle cells and macrophages in the fibroatheromas. Interestingly, proteins related to apoptosis—tumor necrosis factor alpha (TNF-α), interleukin-1 (IL-1) and monocyte chemoattractant protein-1 (MCP-1)—were found to be highly expressed in the EAT [26]. Increased volume of EAT is associated with high-risk plaques [27,28,29,30,31], which is discussed more extensively in the next chapters.

**Table 1 ijms-25-03583-t001:** Summary of the most important studies concerning the role of EAT in the pathogenesis of ACS and cardiac remodeling after MI.

First Author	Year	Study Population/Experimental Model	Main Findings	Reference
Moreno-Santos, I.	2019	Patients with ACS (*n* = 29), stable CAD (*n* = 16) or without CAD (*n* = 29)	Reduced expression of NPR-C, UCP1 and PGC1α in EAT of ACS patients. Decreased activation of p38 MAPK pathway in EAT samples from ACS patients.	[32]
Hao, S.	2023	Male Sprague Dawley rats: MI induction (*n* = 6) or sham surgery (*n* = 6); H9C2 cardiomyocytes	After MI, EAT mediates cardiomyocytes’ apoptosis by secretion of CFD, which causes PARP-1 activation.	[33]
Pedicino, D.	2017	Patients with ACS (*n* = 18), SA (*n* = 16) or without CAD (*n* = 13)	Enhanced NLRP3 and pro-IL1β expression in EAT from ACS patients. Many bacterial species were found in EAT samples from ACS and SA patients.	[34]
Parisi, V.	2020	Patients with CCS (*n* = 54) or recent ACS (*n* = 33)	Reduced IL-1ra levels in EAT from ACS patients.	[35]
Pedicino, D.	2022	Patients with ACS (*n* = 32), CCS (*n* = 34) or MVD (*n* = 12)	Higher content of CD31, CHI3L1, CRP, ENG, IL-17, IL-33, MMP-9, MPO, NGAL, RBP-4, RETN, in EAT of ACS patients found in proteome profiling. Perturbation of the TRBV21 in EAT were associated with the first NSTEMI.	[36]
Langheim, S.	2010	Patients with ACS (*n* = 32), stable CAD (*n* = 34) or without CAD (*n* = 23); HUVEC	Increased resistin expression in EAT of ACS patients. Supernatant of cultured EAT obtained from ACS patients increased permeability of endothelial cells in vitro. Greater number of CD68+ cells in was found EAT of ACS patients than stable CAD patients and controls.	[37]
Rachwalik, M.	2014	Patients undergoing CABG with history of MI (*n* = 17) or without previous MI (*n* = 16)	Previous MI was associated with higher resistin content in EAT.	[38]
Hao, S.	2021	Male Sprague Dawley rats: MI induction (*n* = 20) or sham surgery (*n* = 10); H9C2 cardiomyocytes	EAT-CM through miR-134-5p/KAT7/MnSOD/catalase axis and increase in ROS intracellular levels promoted activation of cardiac fibroblasts into myofibroblasts. Knockdown of miR-134-5p limited myocardial fibrosis in vivo.	[39]
Chang, H-X	2017	Sprague Dawley rats (*n* = 82) which underwent MI (with or without EAT removal) or sham surgery	Increased lipolysis of EAT after MI. EAT removal reduced infarct area, enhanced cardiac function, and decreased inflammation after MI.	[40]

Abbreviations: ACS, acute coronary syndrome; CAD, coronary artery disease; CFD, complement factor D; CHI3L1, chitinase 3-like 1; CRP, C-reactive protein; EAT, epicardial adipose tissue; EAT-CM, conditioned media from epicardial adipose tissue; ENG, endoglin; IL-17, interleukin-17; IL-1ra, interleukin-1 receptor antagonist; IL-33, interleukin-33; KAT7, lysine acetyltransferase 7; MAPK, mitogen-activated protein kinase; MI, myocardial infarction; MMP-9, matrix metallopeptidase 9; MNSOD, manganese superoxide dismutase; MPO, myeloperoxidase; NGAL, neutrophil gelatinase-associated lipocalin (lipocalin 2); NLRP3, NLR family pyrin domain containing 3; NPR-C, natriuretic peptide receptor-C; NSTEMI, non-ST-elevation myocardial infarction; PARP-1, poly(ADP-ribose) polymerase 1; PGC1α, peroxisome proliferator-activated receptor gamma coactivator alpha; pro-IL1β, pro-interleukin-1beta; RBP-4, retinol binding protein 4; RETN, resistin; ROS, reactive oxygen species; SA, stable angina; TRBV21, T cell receptor beta variable 21-1; UCP1, uncoupling protein 1.

EAT represents features of brown adipose tissue (BAT), whose activation is linked with cardioprotective effects [41,42]. Natriuretic peptides (NPs), apart from their role in maintaining cardiovascular homeostasis, may also affect adipose tissue. NP receptors (NPR) are expressed in this tissue, especially NPR-A and NPR-C [43]. NPs may induce lipolysis and promote lipid mobilization. These effects are associated with the activation of NPR-A and, consequently, increased intracellular levels of cyclic guanosine monophosphate [44,45]. Interestingly, NPs can also promote browning of white adipose tissue and enhance the expression of brown fat markers including proteins associated with thermogenesis [46]. The thermogenic role of EAT may diminish with age, obesity, or the presence of CAD [33]. Through the loss of this role, the risk of acute coronary syndromes (ACS) may increase. Patients with ACS had lower expression of NPR-C in EAT in comparison to individuals with stable CAD or without CAD. This was associated with decreased phosphorylation of p38 mitogen-activated protein kinase (MAPK), reduced expression of uncoupling protein 1 (UCP1) and peroxisome proliferator-activated receptor gamma coactivator alpha (PGC1α). This might suggest that the p38 MAPK signaling pathway is diminished, which results in lower expression of brown-like fat genes [32]. On the other hand, the p38 MAPK signaling pathway is strongly activated by cytokines and environmental stress, therefore its activation may contribute to inflammation or apoptosis [47]. However, the brown-like fat phenotype of EAT might be related to atherosclerotic plaque stability. Moreover, medications targeting NPR-C might potentially enhance ACS therapy [32]. PGC1α overexpression in adipocytes alleviated metabolic dysfunction in mice fed a high-fat diet, which depended on heme oxygenase 1 (HO-1). This was associated with browning of adipose tissue and reduced inflammation [48]. In obese mice, PGC1α and HO-1 levels were decreased in epicardial fat in comparison to visceral fat. Moreover, pharmacologically enhancing the activity of HO-1-PGC1α was related to improved LV function in mice [49].

The role of EAT appears to be ambiguous, as it can exhibit both protective and destructive properties towards the heart. The analysis of gene expression in EAT of AF patients showed a high abundance of complement factor D (CFD) transcript [50]. In a study on rats with induced MI, CFD expression was found to be higher in EAT than in subcutaneous adipose tissue (SAT). Additionally, the EAT mass in the MI group was increased compared to that in the sham-operated group. Cardiomyocytes treated in vitro with EAT-conditioned medium (EAT-CM) presented higher rates of apoptosis. Further analysis revealed that EAT secretes CFD, which causes poly(ADP-ribose) polymerase 1 overactivation. This induces apoptosis independently of caspase activity. Moreover, in vivo inhibition of CFD activity was associated with a reduced rate of cardiomyocyte apoptosis and decreased myocardial injury, which confirms its aggravating role in MI-related damage [33].

### 2.1. The Role of Inflammation

Inflammation is directly related to the pathogenesis of atherosclerosis and, consequently, obstructive CAD. Numerous cell types, signaling pathways, and molecules play a role in this process. They could constitute potential treatment targets [51]. Interestingly, the inflammatory process contributing to the progression of this disease extends beyond the atherosclerotic plaques and vessels [52]. Adipose tissue, as a potent participant in local and systemic signaling, can also be engaged in inflammation [53]. Additionally, MI being a severe consequence of CAD is inevitably associated with inflammation [54]. Several studies have described the role of adipose tissue inflammation in ACS, with particular emphasis on EAT.

Inflammation of visceral adipose tissue can be assessed using a microscope. Scavenger macrophages surround dying adipocytes and form crown-like structures (CLS), which are histologic hallmarks of inflammatory processes in adipose tissue [55]. Malavazos et al. analyzed the CLS density in EAT samples obtained from patients undergoing CABG or valvular replacement. Unfortunately, due to the small number of included patients it is hard to consider the result as significant. However, it is worth mentioning that one patient with a rapid weight increase and a recent episode of MI had about 100 times higher CLS density than the remaining CABG patients [56]. Unfortunately, the data concerning CLS formation in EAT are currently limited. Inflammatory cell infiltration in EAT from individuals with CAD is higher than in SAT [57]. The accumulation of mast cells, which secrete multiple vasoactive molecules, may contribute to the rupture of vulnerable plaque. [58]. Moreover, mast cells can activate T lymphocytes. CAD is associated with increased infiltration rates of EAT by T lymphocytes (CD3+) and B lymphocytes (CD20+) [57,59]. However, little is known about the impact of lymphocytes located in EAT on ACS development.

Pedicino et al. proved that EAT expression of NOD-like receptor protein 3 (NLRP3) and pro-interleukin-1β (pro-IL-1β) is significantly enhanced in patients with non-ST-segment elevation acute coronary syndrome (NSTE-ACS) in comparison to individuals with stable angina (SA) or control group [34]. NLRP3 detects microbial patterns, danger signals, or cellular stress, which causes NLRP3 inflammasome assembly and activation. This leads to the release of pro-inflammatory cytokines IL-1β and IL-18 in a caspase-1-dependent manner [60]. These data indicate that ACS might be correlated with upregulation of the some inflammasome components in EAT. In another study, NLRP3 protein level in peripheral blood obtained from patients with ACS was higher than in the control group. It positively correlated with the number of atherosclerotic lesions and diseased coronary arteries [61]. Moreover, the expression of NLRP3 in SAT also positively correlated with the severity of coronary atherosclerosis suggesting that NLRP3 inflammasomes in SAT could be engaged in atherogenesis [62].

On the other hand, another study analyzing patients with ACS undergoing urgent CABG due to NSTEMI and patients with CAD referred to an elective CABG showed that both EAT and circulating IL-1β levels did not differ. However, ACS patients had significantly lower levels of both circulating and EAT IL-1 receptor antagonist (IL-1ra). These results could indicate reduced counterregulatory activity of IL-1ra against IL-1β and its pro-inflammatory role. Furthermore, EAT is actively engaged in the local inflammatory process. Interestingly, no correlation was found between IL-1β levels and markers of M1 macrophages in EAT [35]. In contrast, another study showed that the ratio of M1/M2 macrophages in EAT positively correlates with the levels of pro-inflammatory cytokines and the severity of CAD [63]. However, adipocytes could contribute to IL-1β levels as they can be a source of inflammatory cytokines [64]. Moreover, the inflammatory state of EAT is suggested to be connected to the pathogenesis of INOCA (ischemia with non-obstructive coronary artery) due to several possible mechanisms, such as endothelial function damage [65].

Therapies for atherosclerosis, CAD, and ACS could have an impact on EAT. Moreover, processes in EAT may potentially become therapeutic targets in the future. HMG-CoA reductase (HMGCR) inhibitors, known as statins, are traditionally used as cholesterol-lowering agents. However, their intake is associated with pleiotropic effects [66]. An in vivo study demonstrated that statin therapy was related to decreased EAT thickness and inflammatory profile in patients undergoing aortic valve replacement. Moreover, in vitro analysis showed that the reduction in pro-inflammatory cytokines concentrations was greater in EAT than in SAT, which indicates a possible selective impact of statin therapy on EAT [67]. Furthermore, anti-diabetic medications may affect EAT metabolism. Application of SGLT2 inhibitors in rats with induced AF led to reduced EAT inflammation. This was achieved by upregulation in ketone body levels, downregulated expression of acetyl-CoA carboxylase 1, and decreased GAPDH lysine residue malonylation in macrophages [68]. The use of glucagon-like peptide 1 receptor (GLP1R) agonist can decrease inflammation and adipogenesis, enhance free fatty acid oxygenation, and promote browning of adipocytes [14,69]. The use of SGLT2 inhibitors, GLP1R agonists, metformin and pioglitazone was related to decreased EAT quantity [14,70,71]. These mechanisms may partially explain the beneficial impact of anti-diabetic treatment on the cardiovascular system [14].

During analysis of microbiota composition in EAT, Pedicino and colleagues found DNA of 76 bacterial species. The main species in EAT obtained from ACS patients were Cyanobacteria Streptophyta and Proteobacteria Rickettsiales, whereas Moraxellaceae, Pseudomonas, and Bacteroides dominated in the SA group. It shows the potential role of bacteria in ACS pathogenesis. Unfortunately, it is not yet established if and how they participate in this process [34]. In another study, EAT proteome analysis revealed an elevated amount of pro-inflammatory proteins among NSTEMI patients in comparison to individuals with CAD. The former group had a specific T-cell clonal expansion. In the vast majority of these patients, it was the first CV event. In silico analysis led to a hypothesis that antigens derived from the microbiome are possible inducers of instability through antigen-driven immune response. Even though these results require further confirmation they could be an introduction to developing engineered epitope-based therapies [36].

### 2.2. Adipokines

Adipokines are described as bioactive peptides secreted by adipose tissue. These molecules impact adipose tissue by regulating adipogenesis and adipocyte metabolism and affecting migration of immune cells. However, adipokines may also affect targets at the systemic level. They play a part in inflammation and immune response, glucose metabolism, regulation of blood pressure, myocardial contractility, and other various processes [72]. Thus, adipokines are engaged in the pathogenesis of numerous diseases, and EAT, as both a source and potential recipient of these molecules, is also involved in these processes.

Resistin is an adipokine originally linked with insulin resistance, obesity, and diabetes. However, increasing evidence has revealed that its functions are well beyond that [73]. It has been shown that among patients with acute MI, elevated levels of resistin in the plasma serve as a predictor for all-cause mortality [74]. Another study demonstrated that the concentrations of resistin in EAT obtained from patients undergoing CABG are higher in individuals with prior MI [38]. Langheim et al. observed a higher EAT expression of resistin in ACS patients compared to stable CAD patients and controls. This was accompanied by an increased concentration of resistin in the ACS group. Moreover, in EAT obtained from ACS patients, there was the highest percentage of CD68+ cells (macrophages), which had a similar location as the distribution of resistin [37]. In humans, resistin is mainly secreted by mononuclear leukocytes including macrophages [73,75]. Both in ACS and stable CAD, there was enhanced expression of IL-6, plasminogen activator inhibitor-1 (PAI-1), and MCP-1, which indicates a proinflammatory cytokine profile. Examination with the use of an in vitro model of endothelial permeability showed that among adipokines secreted from EAT, the major player causing endothelial damage is resistin. This may lead to a hypothesis that EAT-derived resistin enters coronary arteries and contributes to the ACS initiation by maintaining aberrant permeability of the endothelium [37]. The exact mechanism is not fully understood. However, resistin may enhance the expression of endothelin-1 and adhesion molecules thus promoting endothelial dysfunction [76]. Moreover, factors secreted by adipocytes seem to enhance the secretion of pro-inflammatory cytokines by endothelial cells [77]. Therefore, activation of macrophages or suppression of resistin action might be potential therapeutic targets for ACS patients.

Fatty-acid-binding protein 4 (FABP4) is a lipid chaperone protein. Apart from its intracellular role, FABP4 is secreted from adipocytes in a controlled manner in response to fasting-related signals. It is also associated with atherosclerosis, diabetes, and other metabolic disorders [78,79]. Peeters et al. discovered that elevated FABP4 expression in carotid atherosclerotic plaques is related to instability of the plaque and correlates with adverse cardiovascular events [80]. Patients presented with acute MI had increased serum levels of fatty acid binding protein 4 (FABP4). Among them, the highest values of FABP4 levels had individuals with cardiac arrest due to ventricular fibrillation. The FABP4 expression was observed in adipose tissue, including EAT. Interestingly, the secretion of FABP4 from adipocytes is mediated through the activation of β3-adrenergic receptors. This finding suggests that during MI, sympathetic signaling could induce lipolysis in EAT surrounding the heart, ultimately leading to the release of FABP4, which may be a potential marker of adrenergic overactivation in cardiovascular diseases [81]. FABP4 increases endoplasmic reticulum (ER) stress, which contributes to the apoptosis of cardiomyocytes. Moreover, the application of FABP4 inhibitor alleviates these deleterious effects. Hence, it can be useful in MI treatment [82].

Omentin-1 is an adipokine that exerts many protective properties. It inhibits inflammation and contributes to vasodilatation by modulation of endothelial nitric oxide synthase phosphorylation and nitric oxide production [83,84]. Administration of omentin-1 in mice was associated with decreased infarct size following ischemia/reperfusion injury [85]. EAT levels of omentin-1 are significantly lower in patients with CAD. There was a negative association between the expression of omentin-1 in EAT and local coronary atherosclerosis [86]. Similarly, the expression of omentin-1 was reduced in EAT samples from patients with prior MI [87]. The plasma levels of omentin-1 increase following myocardial ischemia, whereas its expression in left ventricular tissue decreases.

No study analyzed directly the EAT expression of leptin and adiponectin in patients with ACS. However, based on the available data, it is likely that the secretion of these adipokines from the EAT may play a part in the ACS pathogenesis. Leptin is involved in the pathogenesis of cardiovascular metabolic diseases, including obesity, type 2 diabetes, atherosclerosis, and hypertension [88]. It can also affect plasma cholesterol metabolism [89]. EAT leptin expression was found to be increased in patients with CAD. This was accompanied by elevated serum leptin levels. Moreover, an increase in leptin mRNA expression in the EAT was a risk factor for local coronary artery stenosis [90]. Patients with ACS had higher serum leptin levels than individuals with stable angina [91]. Leptin derived from EAT induced myocardial injury in rats with metabolic syndrome independently from the circulating leptin effects. Furthermore, EAT-derived leptin causes induction of the mitochondrial pathway of apoptosis and promotes inflammation in H9C2 cardiomyocytes. These effects are mediated by protein kinase C/reduced nicotinamide adenine dinucleotide phosphate oxidase/reactive oxygen species pathway [92]. Levels of leptin seem to be inversely related to adiponectin, as they are regulated contrarily. The latter has beneficial effects on the cardiovascular system and exerts anti-inflammatory and anti-fibrotic properties [93]. Patients with coronary atherosclerosis had decreased adiponectin expression in epicardial adipose tissue which was associated with elevated expression of inflammatory cytokines [88,94]. Interestingly, elevated serum adiponectin levels in patients with ACS were related to higher rates of cardiovascular events and mortality in the future. An increase in circulating levels of adiponectin due to adiponectin resistance development is one of the suggested explanations. However, the exact mechanism is not established yet [95].

As presented above, there is evidence that adipokines secreted from EAT are involved in atherogenesis and CAD. However, the mechanisms through which they contribute to the pathogenesis of MI require further studies. Due to the presented findings, the secretory role of EAT should be given particular attention, as it may become a potential therapeutic target in the future.

## 3. Impact of Epicardial Adipose Tissue on Post-Infarct Cardiac Remodeling

Following MI, the cardiac muscle in survivors undergoes pathological changes due to adverse alterations in mechanical and neurohormonal conditions. This process is known as cardiac remodeling [96]. It leads to the development of HF which is related to a poor quality of life and significantly increased mortality [97]. Numerous factors can influence myocardial remodeling and EAT seems to play a role in this process. Various studies have analyzed the mechanisms through which EAT impacts cardiac remodeling.

A study using a rat model demonstrated a positive correlation between EAT mass and both the area of myocardial fibrosis and cardiomyocyte size after MI. Additionally, when rat cardiac fibroblasts were cultured in EAT-CM, it was observed that this environment promoted their activation into myofibroblasts, which produce collagen and therefore contribute to cardiac fibrosis. This effect was achieved by miR-134-5p/KAT7/MnSOD/catalase axis leading to an increase in ROS level. Knockdown of miR-134 in vivo relieved myocardial fibrosis and reduced cardiomyocyte size, thus reducing pathological myocardial remodeling. Interestingly, the use of anti-activin-A antibodies prevented an increase in miR-134-5p levels, which may indicate that the presence of this cytokine in EAT-CM induced aberrant changes [39]. Other studies reported that plasma levels of miR-134-5p were significantly increased in patients with acute MI [98]. miR-134-5p can impact oxidative stress and apoptosis in cardiomyocytes via interaction with cAMP-responsive element binding protein 1 (Creb1) or X-linked inhibitor of apoptosis protein (XIAP) [99,100].

Venteclef et al. proved that EAT plays a role in mediating atrial myocardium fibrosis by secreting adipo-fibrokines, with the main agent being activin A, a member of the TGF-β superfamily. In contrast, the secretome obtained from SAT did not have profibrotic effects and the expression of activin A in SAT was significantly lower, which proves metabolic differences between EAT and SAT. Furthermore, EAT-CM and activin A itself enhanced expression of TGF-β1 and TGF-β2 in rat atria, potentially augmenting profibrotic effects. The levels of activin A in EAT secretome were higher in a subset of patients with left ventricular ejection fraction (LVEF) less than 45% [101]. Another study showed that HF patients had increased serum levels of activin A. These levels significantly correlated with NT-pro-BNP levels, decreased cardiac index, and increased LV end-diastolic pressure. This may indicate that activin A may play a role in the development of HF with reduced LVEF and myocardial remodeling after MI [102].

Type 2 diabetes mellitus (DM2) induces disturbances in the secretory profile of EAT. When cardiomyocytes were treated in vitro with EAT-CM obtained from patients with DM2, impaired contractile function was observed. The primary factors identified as responsible for this dysfunction were activin A and angiopoietin-2 [103]. This is a valuable finding because DM2 is related to a 2-4-fold increase in the risk of HF [104]. Furthermore, DM was present in almost 33% of patients hospitalized due to heart failure [105].

Due to anatomical proximity, pathologies involving the myocardium can also affect EAT. Parisi et al. observed a remodeling of EAT at three months following STEMI. In some patients, the thickness of EAT increased, while in others it decreased. An increase in EAT amount was related to intensified myocardial remodeling and lower LVEF. Simultaneously, circulating levels of IL-13, an anti-inflammatory cytokine involved in cardiac regeneration, decreased three months after STEMI. This decrease was negatively associated with an increase in EAT thickness [106]. The downstream pathways of IL-13 involve ERK1/2 and Akt signaling, which possess pro-proliferative and anti-apoptotic capacities in cardiomyocytes [107,108]. Furthermore, increased circulating levels of IL-13 have been associated with the preservation of LVEF after STEMI [106]. The exact mechanisms of how myocardial ischemia contributes to EAT remodeling have not been established yet. We hypothesize that cardiokines, inflammatory cytokines, and extracellular vesicles (EVs) may play a role in this process. Interestingly, EVs secreted under myocardial ischemia/reperfusion may induce endoplasmic reticulum stress in adipocytes [109].

A quantitative proteomic analysis showed significant changes in the expression of 165 proteins in the EAT of patients with ischemic cardiomyopathy (ICM). The altered proteins were mainly related to cardiac structure or metabolism and immune response. Expression of extracellular matrix (ECM) proteins was significantly enhanced, which was accompanied by increased collagen volume. Activation of ERK1/2 seemed to have been involved in the regulation of ECM remodeling in the EAT of ICM patients, which is in contrast to the beneficial roles of this pathway mentioned above [110].

During the embryonic development of the heart, epicardial progenitors that express WT1 factor undergo epithelial-to-mesenchymal transition (EMT) and become epicardium-derived cells. They are relocated to the heart and form different types of cells [111]. Interestingly, MI leads to activation of the fetal program of the epicardium [112]. After MI, WT1+ mesothelial lineage cells can differentiate into EAT. However, this is possible only within a short period following MI. For this process, activation of the insulin-like growth factor 1 (IGF-1) receptor signaling pathway is required [113]. This may indicate a potential negative impact of IGF-1 pathway activity, even though it is rather depicted as cardioprotective [114,115].

An animal study presented that removal of EAT during the MI induction resulted in better cardiac function and lower infarct area following MI. This procedure was also associated with a decrease in leukocyte numbers in the infarcted hearts one month after MI. Notably, there was an increase in the percentage of CD68+ macrophages and a reduction in both, absolute and proportional number of neutrophils, suggesting an alleviation of inflammation. Moreover, the removal of EAT was accompanied by a decrease in the levels of inflammatory cytokines—TNF-α and IL-1β one month after MI. These results may indicate a possible role of surgical EAT removal in post-MI therapy [40].

## 4. Epicardial Adipose Tissue as a Source of Stem Cells for Cardiac Regeneration

The proliferative potential of human cardiomyocytes (CMs) is limited, especially in adults [116]. The application of stem cells seems to be a possible solution to improve cardiac regeneration following MI. Stem cell therapies were examined in a wide range of cardiovascular pathologies, including HF [117,118]. Animal studies showed promising results of embryonic stem cell (ESC) transplantation [119,120]. However, the use of ESCs raises relevant questions concerning the ethical nature of such procedures [121]. Multiple clinical trials regarding stem cells obtained from various sources in MI therapies have yielded inconsistent results. Mesenchymal stem cells (MSCs) are interesting due to their properties, as they can differentiate into cardiomyocytes, smooth muscle cells or endothelial cells. Moreover, cytokines secreted by MSCs have anti-fibrotic and anti-angiogenic properties. They also promote the differentiation of stem cells resident in the heart [122]. A specific subset of MSCs, known as adipose-derived stem cells (ADSCs), has been isolated from EAT.

ADSCs have the potential to differentiate in many types of cells, including cardiomyocytes, endothelial cells and vascular smooth muscle cells [123,124,125]. ADSCs obtained from cardiac adipose tissue presented an immunophenotype similar to mesenchymal stem cells as they were robustly positive for CD105, CD44, CD166, CD29, and CD90 and negative for CD106, CD45, and CD14. Transplantation of cardiac ADSCs in an animal model of MI led to improved cardiac function and reduced scar size. Moreover, these cells can secrete proangiogenic factors and thus promote angiogenesis resulting in higher capillary density in vivo [126]. Similarly, Özkaynak et al. studied the use of EAT-derived MSCs in a MI rabbit model. The intramyocardial injection of these cells led to an increase in EF, which was accompanied by enriched vascular density and decreased scar size in the infarcted area [127].

It has been established that ADSCs isolated from adipose tissue in distinct locations represent variations in functions, differentiation capacity and therapeutic potential in myocardial infarction. Cardiac ADSCs had significantly higher cell density and proliferation activity than ADSCs obtained from subcutaneous, visceral, and subscapular adipose tissue. They presented increased differentiation potential for cardiovascular cells. Moreover, cardiac ADSCs after systemic transfusion were more frequently recruited to the ischemic myocardium, which was related to improved cardiac recovery and enhanced cardiac functions after MI in mice [128]. Epicardial ADSCs had elevated proliferation potential in comparison to pericardial and omental ADSCs. They also presented the highest cardiomyogenic potential [129]. Epicardial ADSCs had significantly higher expression of GATA4 than pericardial and omental ADSCs [129]. GATA4 is a transcription factor which plays an essential role in myocardial morphogenesis [130,131].

Thankam and Agrawal obtained EAT from hyperlipidemic Yucatan microswine. The researchers isolated EAT-derived stem cells and subjected them to ischemia and/or reperfusion in vitro. With the use of single-cell RNA sequencing, they identified 18 different cell clusters, which may indicate that diverse phenotypes are present. Further analysis showed protective phenotypes presenting several mechanisms, which involve: “epigenetic regulation, myocardial homeostasis, cell integrity, and cell cycle, prevention of fibroblast differentiation, differentiation to myocardial lineage, anti-inflammatory responses, prevention of ER-stress, and increasing the energy metabolism” [132]. Another study with a similar design analyzed ribosomal proteins in exosomes from EAT-derived stem cells. Even though many proteins were identified, the data concerning their extracellular functions are limited [133]. Treating cardiac fibroblasts in vitro with extracellular vesicles obtained from EAT-derived stem cells subjected to ischemia resulted in enhanced expression of cardiomyocyte lineage genes, transcription factors specific for cardiomyocytes (GATA-4, Nkx2.5, IRX4, and TBX5) and myofibroblast biomarker (αSMA) with subsequent reduction in fibroblast (vimentin, FSP1, and podoplanin) and cardiac (connexin-43 and troponin-c) biomarker expression. Analysis with mass spectrometry showed that EVs carried proteins related to cardiac regeneration and inflammation. LGALS1, PRDX2, and CCL2 proteins were identified as important mediators engaged in tissue regeneration [134].

## 5. Imaging Assessment of Epicardial Adipose Tissue and Its Predictive Role

EAT parameters can be assessed using various imaging methods: echocardiography, computed tomography (CT) or magnetic resonance imaging (MRI) (Figure 3). Positron emission tomography (PET) plays a role in the evaluation of EAT metabolic activity. Each of them has advantages and disadvantages which indicate their utility in clinical practice. As EAT is considered as a biomarker of cardiovascular diseases [135,136,137] or a factor indicating future outcomes of patients [138,139], its assessment may become an important element in future diagnostic and therapeutic approaches. The clinical role of EAT is discussed in this chapter.

### 5.1. Methods for Imaging Assessment of EAT

#### 5.1.1. Echocardiography

Echocardiography is a widely performed examination, primarily due to its cost-effectiveness and non-invasive nature, advantages that it holds over magnetic MRI and CT [140]. The measurement is done in transthoracic echocardiography on the free wall of the right ventricle from the parasternal long axis and short axis at end-systole or end-diastole. That region was shown to contain the highest EAT thickness and it allows optimal cursor beam orientation in each view. EAT appears as an echo-lucent space between the external wall of the myocardium and the visceral pericardium [6,141,142]. However, despite its vast availability, it is unable to measure whole epicardial fat and there are conflicting results on the reproducibility of the method [136,142,143].

#### 5.1.2. Computed Tomography

CT enables the measurement of EAT thickness and volume. Additionally, attenuation (density) can be measured, and its less negative values serve as a marker of inflammation within the tissue [144]. Moreover, it grants an assessment of these parameters in the pericoronary adipose tissue (PCAT), which is the fat layer surrounding coronary vessels [145]. Altogether, these make CT more multi-functional in EAT evaluation than echocardiography. Nevertheless, it is numbered among the more expensive methods.

#### 5.1.3. Magnetic Resonance Imaging

MRI is stated as the best method to assess visceral fat depots and the EAT amongst them. Contrary to CT imaging it is a radiation-free method with high resolution allowing distinct exact borders of EAT as well as visualize it in many small cardiac spaces for example superior interventricular groove [146]. However, this approach is more expensive and time-consuming than CT and echocardiography.

#### 5.1.4. Positron Emission Tomography

It has been shown that fluorodeoxyglucose (FDG) uptake visualized by PET corresponds to macrophage infiltration within the tissue, therefore it can be used to assess the inflammatory properties of PCAT [147]. PET imaging is usually combined with CT (PET/CT), which allows both metabolic and anatomical evaluation.

### 5.2. EAT Parameters Assessed in Imaging Studies

#### 5.2.1. EAT Thickness

EAT thickness measured in echocardiography was associated with some CAD risk factors such as abdominal visceral adipose tissue, C-reactive protein level, waist circumference, age, and body mass index (BMI) [148]. A meta-analysis of nearly 5000 patients showed that EAT thickness is related to CAD and its severity [149]. Other studies also revealed EAT thickness measured in echocardiography to be a predictor of CAD [150,151] as well as an independent factor affecting its severity [152,153,154] and it was moreover reported to foresee the presence of multivessel disease [155]. Multiple studies identified EAT thickness measured in echocardiography to parallel with the Gensini score [151,155,156] and SYNTAX score [151,157], also in patients with ACS [137,156]. In addition, EAT thickness correlates with TIMI flow grade which is used for evaluation of coronary perfusion [158]. Moreover, a robust relationship with the number of thin-cap fibroatheromas was reported. Patients with thick EAT in echocardiography had approximately more and larger plaques, they were of higher mean burden index, with bigger necrotic core [159] and contained more lipids [160]. In a study on asymptomatic patients with type 2 diabetes, EAT thickness measured by MRI in the left atrioventricular groove revealed thicker EAT in patients with more severe coronary stenosis, yet no significant relation was showed between EAT thickness and silent myocardial ischemia [161].

The EAT thickness measured in CT showed a correlation with coronary artery calcium score (CAC), predicted CAD in asymptomatic patients [16] and negatively correlated with impairment in myocardial flow reserve [162]. Nonetheless, when combined with the calcium scoring, it gained little improvement in coronary stenosis diagnosis [163]. Picard et al. reported the EAT thickness on the free wall of the left ventricle correlated with the presence and severity of CAD. No such relation was detected for the right ventricle. The authors claim that the method has low diagnostic value compared to calcium scoring and CT coronary angiography [163].

Echocardiographic EAT thickness correlated weakly with the Framingham risk score, which is an algorithm used to estimate 10-year cardiovascular risk [164]. Tanindi et al. proved that EAT thickness might be used as a powerful predictor of acute MI. Echocardiographic measurement of EAT may be a useful tool when deciding about aggressive therapy or as a follow-up parameter [165,166]. It also appeared to be an independent risk factor of NSTEMI, unstable angina pectoris [167], and restenosis after PCI [168]. Additionally, the EAT thickness in echocardiography occurred to be significantly predictive of adverse cardiovascular events in patients presenting with AF [169] and ACS during short-term and long-term follow-up [170,171]. Moreover, individuals with high echocardiographic EAT thickness were more likely to require target vessel revascularization. However, there was no significant dependence when it came to recurrent MI [172]. On the other hand, EAT thickness measured in CT in right, left, and anterior interventricular fossae predicted MI in patients with COVID-19 [173]. In a meta-analysis of over 6600 cases EAT thickness measured in either CT or echocardiography was proven to be increased in patients with myocardial infarction (MI) [136].

Nevertheless, the ability of echocardiographic EAT thickness to predict major adverse cardiovascular events (MACE) is denied by some results [174,175], especially in patients undergoing hemodialysis [176]. Some works deny the existence of any interaction with GRACE score and TIMI score, which are used to estimate the likelihood of adverse cardiac events [156,177]. Nonetheless, the latter was indicated by other researchers [167,178].

The EAT thickness measured in transthoracic echocardiography in subjects with MI corresponded positively with ST—segment resolution, reduction in LVEF, reduction in LV end-diastolic and end-systolic volume, and larger infarct size [171,179]. Furthermore, thick EAT was correlated with the presence of coronary collateral vessels (CCVs) in patients presenting with ACS. It suggests that EAT plays a role in the development of CCVs, which may contribute to alleviating ACS outcomes [180]. Excess echocardiographic EAT thickness was an independent predictor of no-reflow phenomenon in STEMI patients who underwent PCI [179,181], and coronary slow flow phenomenon, which appears as a slowed stream of contrast without visible stenosis in coronary angiography. It is an indicator of endothelial dysfunction and the early phase of atherosclerosis [182]. In addition, in STEMI subjects after PCI thick EAT assessed by echocardiography predicted the new onset of atrial fibrillation (AF) during hospital follow-up which means that the inflammatory properties of EAT may take a role in atrial myocardial remodeling [137].

#### 5.2.2. EAT Volume

High EAT volume evaluated in CT showed a correlation with the presence of ACS [183], thin-capped fibroatheroma and high lipid content in the plaque [184]. Both ROMICAT trials also revealed the discriminative role of EAT volume assessed in CT for discovering high-risk lesions [185,186]. Excess EAT volume in CT correlated with vulnerable plaque characteristics, such as low density, presence of remodeling, and non-calcified component. However, there were weak associations with the amount of plaque [27,28,29,30,31].

The EAT volume and EAT volume indexed to the body surface (EATvi) measured in CT showed a correlation with CAC score and was associated with CAD [187,188,189,190,191,192]—this was also the case in patients with diabetes [193] and HIV [194]. High EAT volume evaluated in CT positively corresponded with CAC score in three main coronary arteries [195] and negatively correlated with impairment in myocardial flow reserve but was weaker than EAT thickness [162]. EAT volume measured in CT significantly correlated with the severity of CAD [196], but not with the severity of stenoses [189], and when combined with the calcium scoring it gained little improvement in coronary stenosis diagnosis [197]. Nonetheless, Milanese et al. denied the predictive role of EAT volume measured in CT regarding CAD [198]. Additionally, Goeller et al. showed in a population of patients without diagnosed CAD, that EAT volume evaluated in CT was higher in individuals with early atherosclerosis compared to those with more advanced disease. There was no correlation between EAT volume and CAC score value. Moreover, EAT volume was a strong predictor of cardiac death and MI [139].

The Framingham Heart Study and The Heinz Nixdorf Recall Study presented a significant relation between EAT volume measured in CT and the prevalence of fatal and non-fatal coronary events [138,199]. This is supported by many trials which confirmed that increased EAT volume evaluated in CT, EATvi, and EAT indexed to the BMI are independent predictors of MI, cardiac death, and other coronary events [195,200,201,202,203], amongst patients with type 2 diabetes [204] and those with HIV [205]. EAT volume assessed in CT was positively associated with the prevalence of myocardial ischemia in diabetic patients. However, it did not correlate with its severity and there are conflicting results regarding the comparison of EAT and the CAC score performances [206,207,208]. Furthermore, EAT volume measured in CT was an independent factor related to the presence of coronary slow-flow phenomenon [209]. Nonetheless, no relation was observed between EAT volume and either fractional flow reserve (FFR) [210], or CT perfusion imaging [211].

Recently, AI has attracted much interest in the field of medical studies, including cardiology [212,213]. Deep learning (DL) models can be used to automatically quantify EAT from coronary CT angiography images [195]. Commandeur et al. used the machine learning (ML) model to predict the long-term risk of MI and cardiac death in asymptomatic subjects. Apart from many clinical parameters, the algorithm used CAC and EAT volume. The latter was calculated automatically with the use of a DL-based method. The ML model significantly better predicted the risks than the atherosclerotic cardiovascular disease risk score or CAC score alone [214]. In another study with a similar design, apart from clinical and imaging features, serum concentrations of biomarkers were also implemented in the ML model. This resulted in superior prediction of CV events than contemporary tools used for risk assessment [215].

In a retrospective analysis, Fisser et al. showed that EAT volume measured in MRI positively correlated with prevalence of MI, number of vessels obstructed, and amplitude of ST-elevation. However, there was no association with myocardial salvage index which is a tool presenting the treatment efficacy [216]. On the other hand, multiple studies exposed that more excess EAT volume evaluated in MRI goes along with smaller infarct size and lower number of vessels obstructed [217]. Nevertheless, there was still a positive correlation between EAT volume and myocardial salvage index [218]. A cross-sectional UK Biobank cardiovascular magnetic resonance imaging substudy acknowledged that CAD or history of MI did not correspond with the EATvi [219]. Homsi et al. demonstrated that more excess volume of EAT assessed in MRI predicted myocardial infarction in hypertensive men, yet there was no correlation with CAD or number of affected vessels [220].

#### 5.2.3. EAT Attenuation and Mass

EAT attenuation measured in CT negatively correlated with CAC score [221] and CAC score in each of the three main coronary arteries, but not regarding the left circumflex coronary artery (LCx) [195]. In another study, there was no significant relation between the severity of CAD [196] and the presence of high-risk lesions in coronary arteries [186].

Patients presenting with MI had significantly higher EAT attenuation assessed in CT [202]. However, another study showed no significant difference between ischemic or non-ischemic lesions and EAT attenuation [222]. Goeller et al. presented that EAT attenuation measured in CT was relevantly lower in patients with early atherosclerosis when compared to ones with more advanced disease. In his study, no correlation was shown with the CAC score, yet EAT attenuation was a better predictor than EAT volume of cardiac death and MI [139]. Nonetheless, another study exhibited EAT attenuation to be a weaker predictor of myocardial ischemia and CAC score than EAT volume [208]. EAT attenuation evaluated in CT independently predicted long-term MACE in asymptomatic patients [195].

In patients with heart failure, the whole EAT mass evaluated in MRI was significantly lower than in healthy controls [223]. Moreover, it was lower in individuals with reduced LVEF separately from CAD. It indicates that the correlation between EAT mass and CAD severity is highly dependent on LVEF [224]. Furthermore, it was revealed that EAT mass (assessed in MRI) indexed to the body surface lower than 22 g/m2 was a predictor of cardiac death [225].

#### 5.2.4. Pericoronary Adipose Tissue (PCAT)

Pericoronary adipose tissue is part of EAT which stays in direct contact with coronary vessels thus it has a paracrine effect on them. It can be measured using CT and cardiac MRI. Therefore, this subchapter describes associations of PCAT volume or attenuation with coronary arteries and cardiovascular outcomes [226].

In the CRISP-CT study, the authors revealed a relevant association between high fat attenuation index around LAD and RCA, and prospective cardiac mortality risk. However, it was diminished amongst patients who had been put on statins or aspirin therapy and there was no such correlation regarding LCx [227]. Combining PCAT volume measured in CT around RCA, LAD, and LCx showed statistically significant potential in distinguishing NSTEMI from unstable angina pectoris in contrast to EAT volume [228]. RCA-PCAT attenuation assessed in CT positively correlated with total atherosclerotic plaque burden and amount of calcified plaque, however, there was no relationship regarding the prevalence of MI, cardiac death or revascularization [229]. PCAT volume was found to be significantly associated with the presence of a culprit lesion in the artery. However, such a relation was absent for PCAT attenuation [230]. In another study, PCAT attenuation, but not PCAT volume, was markedly higher in patients with acute MI compared to individuals with stable CAD [196].

Shan et al. showed that only LAD-PCAT volume evaluated in CT correlated with myocardial ischemia measured by fractional flow reserve (FFR) and when combined with CAD-RADS categories it gave a huge improvement in the prediction of myocardial ischemia [210]. In contrast, Duncker et al. showed that only RCA-PCAT attenuation and average PCAT attenuation, both measured in CT were significantly related to myocardial ischemia assessed by FFR, which was false for every other PCAT volume and its attenuation as well as EAT and paracardial adipose tissue volume and attenuation [231]. Zhou et al. demonstrated that combining radiomics features of PCAT evaluated using CT with or without plaque characteristics was significantly related to FFR. However, plaque characteristics themselves gave poor performance. The authors conclude that PCAT characteristics may be of better predictive value regarding myocardial ischemia than the whole EAT [232].

Toya et al. presented the predictive ability of PCAT measured in MRI regarding MACE, especially those around the superior interventricular groove [233]. Amongst PCAT thickness indexed to the body surface, only the one measured in the superior interventricular groove using MRI showed statistical significance in prediction of MACE after STEMI [234].

Mazurek et al. presented nine patients with non-ST segment elevation ACS with PCAT showing higher FDG uptake evaluated in PET/CT than other fat depots, which indicates its pro-inflammatory properties. FDG uptake was also related to negative plaque characteristics and revealed a positive correlation with plaque burden and necrotic core rate as well as a negative correlation with fibrous plaque rate [147]. Furthermore, the authors exhibited that PCAT FDG uptake significantly paralleled with CAD presence. In overweight patients, it positively correlated with the percent of coronary stenosis indicating that weight gain influences PCAT inflammatory activity. Moreover, PCAT FDG uptake was an independent predictor of stenoses in RCA and LAD [235]. In another study, PCAT FDG uptake was higher in patients with vasospastic angina (VSA) in LAD and positively corresponded with the density of vasa vasorum in the artery. It suggests that VSA has an inflammatory background related to the activity of PCAT [236].

## 6. Conclusions

Numerous studies have confirmed that EAT contributes to the pathogenesis of CAD and can be useful in the prediction of ACS. EAT evaluation during routine imaging studies like echocardiography and CT could be potentially implemented in risk assessment scores. Additionally, EAT plays a role in cardiac remodeling following MI, primarily by promoting fibrosis, and medications that reduce the secretion of fibrokines from EAT may mitigate this adverse cardiac remodeling. Conversely, EAT can exhibit cardioprotective properties and can be a source of SCs capable of enhancing myocardial regeneration after MI. Therefore, we need significantly more research on the dual role of EAT before we can proceed with its application in clinical practice.

## Figures and Tables

**Figure 1 ijms-25-03583-f001:**
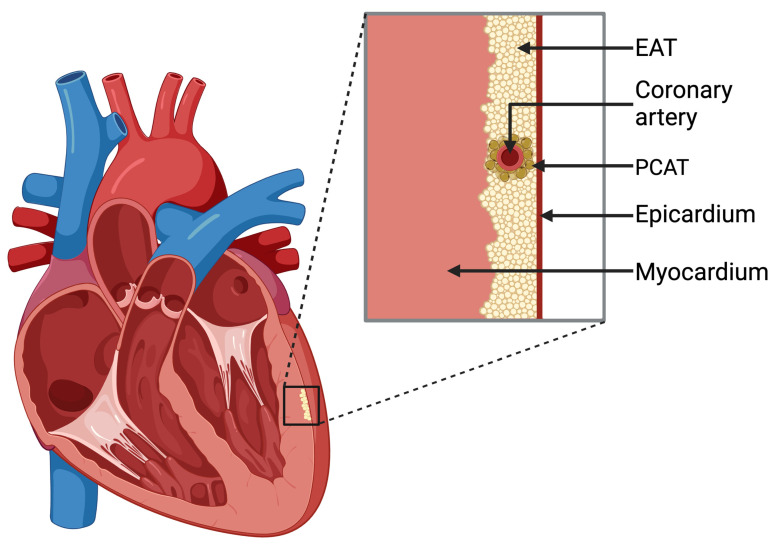
Schematic of the epicardial adipose tissue (EAT) and pericoronary adipose tissue (PCAT) locations within the heart wall. Created with BioRender.com (accessed on 13 February 2024).

**Figure 2 ijms-25-03583-f002:**
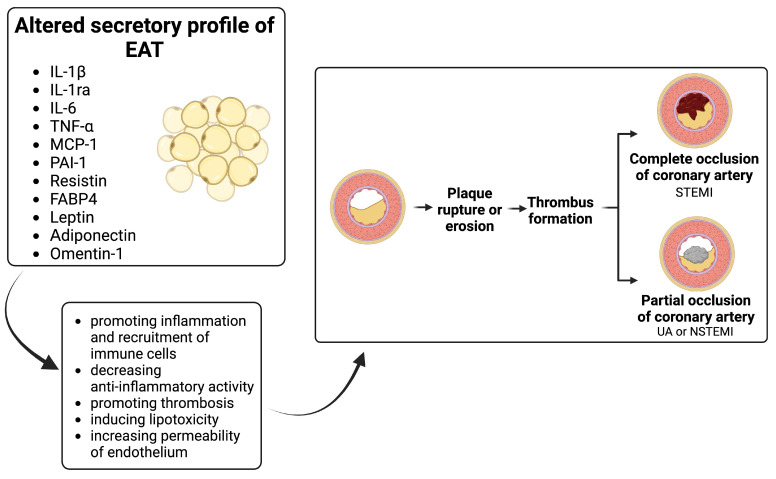
Altered secretory profile of epicardial adipose tissue and its impact on the pathogenesis of acute coronary syndrome with proposed mechanisms. Abbreviations: FABP4, fatty acid binding protein 4; IL-1ra, interleukin-1 receptor antagonist; IL-1β, interleukin-1β; IL-6, interleukin-6, MCP-1, monocyte chemoattractant protein-1; NSTEMI, non-ST-elevation myocardial infarction; PAI-1, plasminogen activator inhibitor-1; STEMI, ST-elevation myocardial infarction; TNF-α, tumor necrosis factor alpha; UA, unstable angina. Created with BioRender.com (accessed on 13 February 2024).

**Figure 3 ijms-25-03583-f003:**
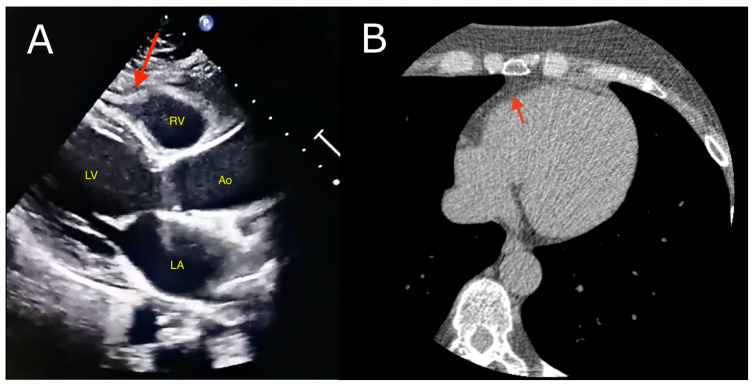
Epicardial adipose tissue (red arrows) visualized in: (**A**) transthoracic echocardiography—parasternal long axis view, (**B**) computed tomography. Abbreviations: Ao, aorta; LA, left atrium; LV, left ventricle; RV, right ventricle.

## Data Availability

Not applicable. No new data were created.

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
