# Peer review of "The Role of Epicardial Adipose Tissue in Acute Coronary Syndromes, Post-Infarct Remodeling and Cardiac Regeneration"

_ijms, 2024, doi:10.3390/ijms25073583_

Round 1
Reviewer 1 Report
Comments and Suggestions for Authors
The paper is well written. I suggest to improve discussion of the role of epicardial adipose tissue on combined PET/CT imaging
Comments on the Quality of English LanguageMinor english revision
Author Response
Dear Reviewer,
Thank you for taking the time to review our manuscript. We appreciate the positive feedback. In response to your comment, we extended the discussion by adding information about PET/CT imaging in EAT assessment (lines 448-449, 486-490, 666-677). We also performed English language editing. All the changes have been highlighted in yellow.
Sincerely,
M.Wojciechowska
Reviewer 2 Report
Comments and Suggestions for Authors
Comments to the author:
The Manuscript " The Role of Epicardial Adipose Tissue in Acute Coronary Syndromes, Post-infarct Remodeling and Cardiac Regeneration. The manuscript describes about imaging assessment of EAT, its prognostic value and correlation with clinical characteristics of patients
Revisions required:
1- Pericardial and epicardial fats are different please differentiate role of both kind of fats differently
2- Please discuss role of PGC 1 alpha, HO1 I in the context of pericardial and epicardial fat
Comments on the Quality of English LanguageMinor english corrections are required
Author Response
Dear Reviewer,
Thank you for taking the time to review our manuscript. We appreciate the thorough feedback and constructive criticism provided. In response to your comments, we have made the following revisions to the manuscript:
- We strengthened the information that pericardial adipose tissue and epicardial adipose tissue are different fat depots (lines 38-40). As the article focuses on the role of EAT, we opted not to dedicate separate sections to pericardial fat. However, we deleted some phrases that could be misleading (lines 603-605). We recognize the importance of pericardial fat and suggest it as a potential topic for future exploration. Thank you for your consideration.
- We added the discussion about the role of PGC 1 alpha, HO1 in the context of epicardial fat (lines 127-132)
We also performed English language editing.
All the changes have been highlighted in yellow.
Sincerely,
- Wociechowska

Reviewer 3 Report
Comments and Suggestions for Authors
This is an interesting review dealing on crosstalk between the heart and epicardial adipose tissue in the context of post-MI remodeling and regeneration.
Few questions:
How is the communication between EAT and myocardium mediated? Diffusion, circulation…?
Is there a potential mechanism described how natriuretic peptides regulate EAT morphology and function?
Is there any data demonstrating impact of anti-diabetic treatment and the EAT size?
On the one hand it is perceivable that changes in EAT affect cardiac function, on the other, how would myocardial changes affect EAT? Arte there possible explanations for this?
What about applying PET as an imaging method to assess EAT?
Author Response
Dear Reviewer,
Thank you for taking the time to review our manuscript. We really appreciate the meticulous feedback and valuable questions, which has helped improve the quality of our work. In response to your questions, we have made the following revisions to the manuscript:
- We specified information about communication between the EAT and the myocardium (lines 49-50).
- We added information about natriuretic peptides and their possible impact on EAT (lines 108-115).
- We included information about the impact of anti-diabetic treatment on the EAT size (lines 206, 210-215 ).
- We added some possible explanations of how the myocardium may affect the EAT (lines 361-365). Unfortunately, we found that the knowledge concerning this issue is very limited.
- We extended the discussion by adding information about PET/CT imaging in EAT assessment (lines 448-449, 486-490, 666-677).
All the changes have been highlighted in yellow.
Once again, we would like to express our gratitude for your guidance.
Sincerely,
M. Wojciechowska

Reviewer 4 Report
Comments and Suggestions for Authors
The authors present the review devoted to the pathophysiology of EAT, its interconnection with the risk of MACE, including myocardial infarction and means of visualization of EAT. Non-standard approach was undertaken in description of measuring of EAT thickness and EAT volume, as authors described how these measurement are interconnected with cardiovascular risk, being measured by various techniques.
The review is supplemented with 2 Figures. Includes 215 references, the majority of which represent state of the art results.
I congratulate the authors with the successful work. I have a few recommendations which could have improved the article.
1. In Adipokines chapter the review of leptin and adiponectin roles in EAT dysfunction and ACS may be added
2. Inflammation chapter misses information of the adaptive immune cells, their presence in EAT and possible implications in ACS development.
3. The table summarizing major findings related to the role of EAT in ACS could have improved the perception of the material.
Author Response
Dear Reviewer,
Thank you for taking the time to review our manuscript. Your insightful feedback was unquestionably helpful in correcting our article. In response to your recommendations, we have made the following revisions to the manuscript:
- We reviewed the roles of leptin and adiponectin (lines 285-307).
- We added information about adaptive immune cells (lines 165-170). Unfortunately, we realized that the knowledge concerning this issue is very limited. We did not find any study directly analyzing the role of lymphocytes located in EAT in the context of ACS. Thus we were unable to elaborate on this topic.
- We added a table summarizing the most important studies concerning the role of EAT in the pathogenesis of ACS and cardiac remodeling after MI (Table 1, Pages 4-5).
All the changes have been highlighted in yellow.
We wish to express our sincere appreciation for your invaluable support.
Sincerely,
M. Wojciechowska
